# Transcriptional synergy as an emergent property defining cell subpopulation identity enables population shift

Satoshi Okawa[1], Carmen Saltó[2], Srikanth Ravichandran[1], Shanzheng Yang [2], Enrique M. Toledo [2,4], Ernest Arenas [2] & Antonio del Sol[1,3]

Single-cell RNA sequencing allows defining molecularly distinct cell subpopulations. However, the identification of specific sets of transcription factors (TFs) that define the identity of these subpopulations remains a challenge. Here we propose that subpopulation identity emerges from the synergistic activity of multiple TFs. Based on this concept, we develop a computational platform (TransSyn) for identifying synergistic transcriptional cores that determine cell subpopulation identities. TransSyn leverages single-cell RNA-seq data, and performs a dynamic search for an optimal synergistic transcriptional core using an information theoretic measure of synergy. A large-scale TransSyn analysis identifies transcriptional cores for 186 subpopulations, and predicts identity conversion TFs between 3786 pairs of cell subpopulations. Finally, TransSyn predictions enable experimental conversion of human hindbrain neuroepithelial cells into medial floor plate midbrain progenitors, capable of rapidly differentiating into dopaminergic neurons. Thus, TransSyn can facilitate designing strategies for conversion of cell subpopulation identities with potential applications in regenerative medicine.

---

[1] Computational Biology Group, Luxembourg Centre for Systems Biomedicine (LCSB), University of Luxembourg, 6, avenue du Swing, L-4367 Belvaux, Luxembourg. [2] Department of Medical Biochemistry and Biophysics, Laboratory of Molecular Neurobiology, Biomedicum 6C, Solnavägen 9, Karolinska Institutet, 17177 Stockholm, Sweden. [3] Moscow Institute of Physics and Technology, Dolgoprudny 141701, Russia. [4] Present address: Novo Nordisk Research Centre Oxford (NNRCO), Cellular and Systems Genomics, Oxford OX3 7BN, United Kingdom. These authors contributed equally: Satoshi Okawa, Carmen Saltó. Correspondence and requests for materials should be addressed to A.D.S. (email: antonio.delsol@uni.lu)

Recent advances in single-cell RNA-seq technologies have allowed to classify cells into distinct cell subpopulations based on their gene expression profiles. The identity of these cell subpopulations can range from well-defined cell types, subtypes of a same cell type to cells with unclear characters. It has been observed that a handful of specific TFs is sufficient to maintain cell subpopulation identity[1]. Identification of such core TFs can facilitate the characterization and conversion of any cell subpopulation, including rare and previously unknown ones, opening thus novel functional applications[2]. However, this is a challenge since the core TFs that determine the identity of such novel cell subpopulations are largely unknown. Importantly, the definition of identity TFs is dependent on the cellular context in which it is employed[3]. In the context of cell/tissue types, for example between neurons and hepatocytes, the identity TFs are defined by the comparison between these largely different cell types. However, in the context of cell subpopulations within a cell type, such as different subtypes of dopaminergic neurons[4], the definition of identity TFs becomes subtler due to the increased commonality between them.

Existing methods for identifying TFs for cell identity or cellular conversions[5–7] rely on a set of gene expression profiles of bulk cell/tissue types. Consequently, the application of these methods is limited to those bulk cell/tissue types, and cannot be applied to novel subpopulations of cells identified in a newly generated single-cell dataset. In addition, these methods detect potential identity TFs by focusing on properties of individual TFs, such as gene expression levels or the number of their unique target genes, rather than emergent properties of potential identity TFs themselves, such as transcriptional synergy among them.

Combinatorial binding of specific TFs to enhancers is known to result in a synergistic activity essential for robust and specific transcriptional programmes during development[8]. The functionality of several TFs operating together to achieve a common output has been studied in detail in embryonic stem cells (ESCs), where a transcriptional core involving Pou5f1, Sox2, and Nanog controls pluripotency[9]. Furthermore, it has been observed in different systems that multiple TFs are required to function cooperatively to sustain the overall cellular phenotype[10].

Here, we propose the general concept that cell subpopulation identity is an emergent property arising from a synergistic activity of multiple TFs that stabilizes their gene expression levels. Based on this concept, we develop a computational platform, TransSyn, for the identification of synergistic transcriptional cores defining cell subpopulation identities. TransSyn does not depend on the inference of gene regulatory networks (GRNs), which are often incomplete and their topological characteristics not always capture the multiple direct and indirect interactions between genes. In addition, it only requires a single-cell RNA-seq data of distinct subpopulations as input (Fig. 1a), and does not depend on pre-compiled gene expression datasets or any other prior knowledge. Consequently, TransSyn infers subpopulation identities within a cell population, and aids in designing strategies to convert cell subpopulation identities, especially in cases of closely related subpopulations in functionally different states. Finally, as a direct application of TransSyn, we show that the knowledge of cell subpopulation-specific synergistic transcriptional cores enables experimental conversion of human hindbrain neuroepithelial cells into medial floor plate midbrain progenitors, which rapidly differentiate into DA neurons. Thus, TransSyn can facilitate conversion of cell subpopulation identities with potential applications in regenerative medicine.

## Results

### Rationale and outline of the method.

TransSyn identifies a specific combination of TFs that are most frequently expressed and exhibit high transcriptional synergy computed by multivariate mutual information (MMI)[11]. MMI measures the information (i.e., predictability) gained by an additional variable (TF), which cannot be explained by the simple summation of the information given by the subsets of variables. For example, MMI among three TFs, $X$, $Y$, and $Z$, is defined as:

$$\text{MMI}(X; Y; Z) = I(X; Z) + I(Y; Z) - I(X, Y; Z),$$

This indicates that when MMI is negative, the three TFs are synergistically interacting with each other, because the knowledge of both $X$ and $Y$ together (i.e., $I(X, Y; Z)$) provides more information about Z than the sum of the knowledges given by $X$ and $Y$ separately (i.e., $I(X; Z)+I(Y; Z)$) (Fig. 1b). The same principle applies to MMI with higher numbers of variables. In this way, TransSyn considers all possible direct and indirect regulatory interactions that can be measured by gene expression. Therefore, it can account for the disparate nature of synergistic transcriptional regulation, including combinatorial/cooperative binding of TFs to target gene promoter/enhancer regions[8], and protein-protein interactions among transcriptional co-factors.

TransSyn requires single-cell RNA-seq data for MMI computation. Ideally, MMI for all possible combinations of TFs should be calculated to identify the most synergistic TF combination. However, such computation is infeasible (for example, the number of all combinations of 3, 4, 5, and 6 TFs among 100 TFs already adds up to 1, 271, 422, 845). Therefore, we implemented a dynamic search algorithm, in which an initial set of most synergistic 3-TF combinations (seed combinations) are progressively extended by adding TFs one by one as long as MMI calculated for the new combination exceeds the MMI of the previous seed combination (Fig. 1c; Supplementary Fig. 1) (see Methods). The search is terminated when the addition of a new TF results in no further decrease in MMI, and the current TF combination exhibiting the least MMI (i.e., most synergistic) is considered the synergistic transcriptional core. Upon termination, if more than one TF combination exhibits the highest synergy, they are ranked by another information theoretic measurement, total correlation (TC), which, unlike MMI, incorporates interactions between all possible combinations of TFs within each core providing a measure of interaction strength[12].

**TransSyn captures known synergistic transcriptional cores.** By applying TransSyn to a large compilation of published single-cell RNA-seq data, we created a catalog of synergistic transcriptional cores specific to 186 cell subpopulations (Supplementary Data 1). Here, by subpopulations we mean distinct groups of cells within a heterogeneous cell population identified based on their gene expression profiles, and do not discriminate between well-defined cell types, subtypes of a same cell type and cells with unclear identity. The predicted synergistic transcriptional cores, when evidence is available, consistently contained TFs known to maintain the respective cell subpopulation identities. For example, the key pluripotency factors POU5F1, NANOG, and SOX2 that maintain the ESC phenotype were found as the most synergistic transcriptional core in hESCs (Table 1; Supplementary Data 1). Notably, these TFs have been speculated to act synergistically via large clusters of enhancers[13]. Another example is the blood progenitor subpopulation[14] that contained Tal1, Gata2, Runx1, and Fli1 in its synergistic transcriptional core (Table 1; Supplementary Data 1). These TFs have been shown to form complexes via protein-protein interactions that stabilize their cooperative binding to DNA and synergistically control the subpopulation identity[15]. Therefore, this represents another known example where a synergistic interaction of TFs defines a cell subpopulation identity. Finally, the synergistic core of human

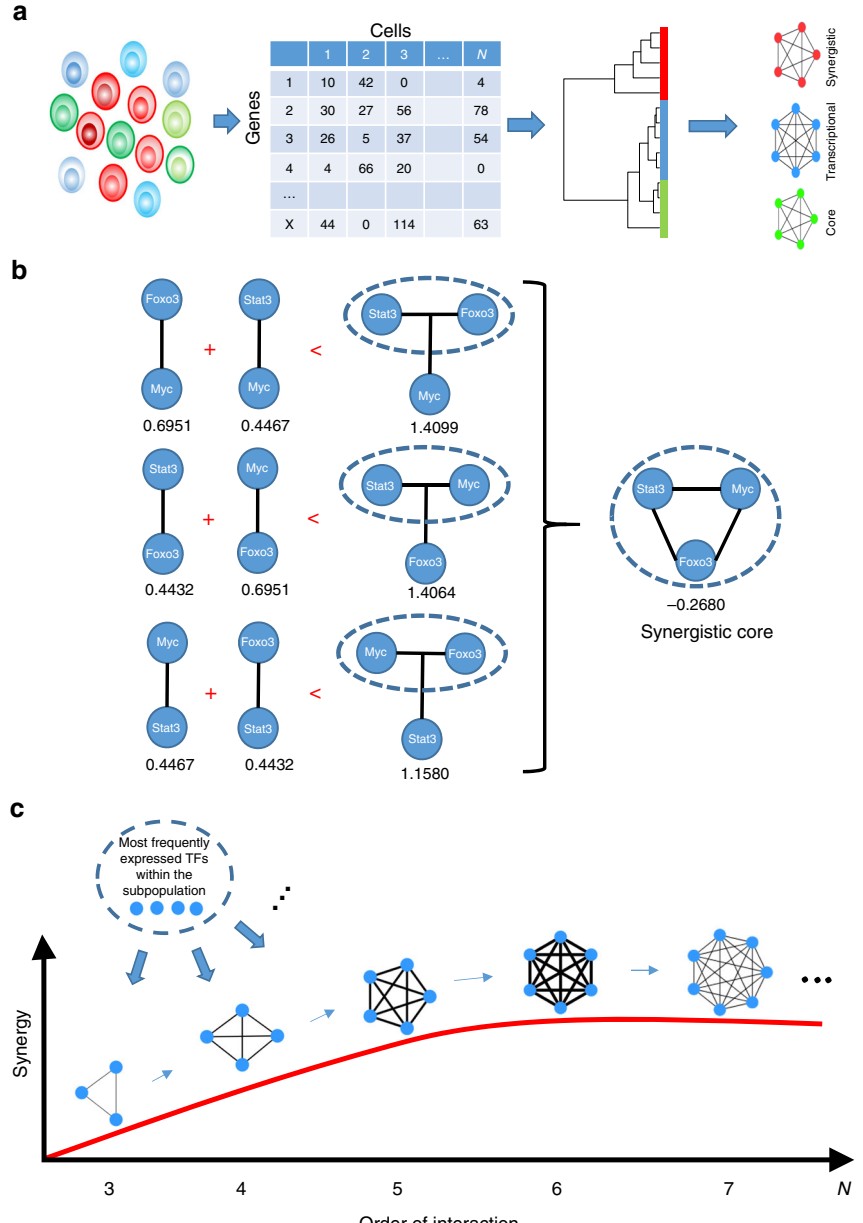

**Fig. 1** Principle of transcriptional synergy and method overview. **a** The method requires single-cell RNA-seq data classified into distinct subpopulations as input and identifies most synergistic transcriptional cores for each subpopulation. **b** Comparison of pair-wise MI between individual TF pairs with joint MI between two TFs together and a third one. For a combination of TFs to be synergistic, the sum of pair-wise MIs has to be less than the joint MI (i.e., negative MMI). Any permutation of same set of TFs results in the same MMI value. **c** Dynamic search for identifying the most synergistic transcriptional cores, in which the "seed" 3-TF combinations are progressively expanded by an addition of another TF one by one. The search is terminated when there is no more increase in synergy when adding a new TF to the current best combination and the current combination is considered the most synergistic transcriptional core

fetal oculomotor and trochlear nucleus (hOMTN) subpopulation consisted of ISL1 and PHOX2A (Supplementary Data 1), which have been shown to synergistically specify cranial motor neurons from mESCs[16].

TransSyn predictions also contained several TFs known to interact with each other to control cell subpopulation identities. For example, Gata1, Gfi1b Klf1, and Ikzf1, known to maintain embryonic blood cells[17, 18] were found in the synergistic transcriptional core of the embryonic primitive erythrocyte subpopulation[14] (Supplementary Data 1). Gata1 and Ikzf1 are known to functionally regulate each other. In addition, the synergistic transcriptional core of the embryonic visceral endoderm subpopulation[14] included Eomes, Otx2, Zic3, Foxa2, and Hnf4a (Table 1; Supplementary Data 1), which are known to regulate each other and other downstream targets specific to this cell subpopulation[19, 20]. Id3, Klf13, Klf6, and Klf4 are known for their roles in the acquisition of vascular endothelial cell fate, whose synergistic transcriptional core contained these TFs[21, 22]. The synergistic core of the mouse enteroendocrine cell contained Neurog3, Neurog1, Insm1, Nkx2-2, Foxa1, Foxa2, Pax4, and Lmx1a, all of which are known to be essential for the functioning of the cell[23–29]. We also examined the synergistic transcriptional core of the human subpopulations for which only mouse functional data is available, such as hProgFPM and hDA2

**Table 1 Most synergistic transcriptional cores predicted by TransSyn and top 10 JSD TFs in example subpopulations, where known identity TFs are in bold**

| Data set | Cell subpopulation | Synergistic transcriptional core | Top 10 JSD TFs (incl. ties) |
|---|---|---|---|
| Treutlein et al. 2014 | Lung surfactant-secreting cuboidal alveolar type 2 cell | Atf4, Fos, **Sox9**, Sp3, **Irx1** | Gfi1, Hes7, Insm1, Mesp2, Nr2e1, Phox2a, Sp5, Tox2, Zbtb12, Zfp251, Zfp398, Zkscan16 |
| Grün et al. 2015 | Intestinal organoid enterocyte precursor | **Gata4**, Rxra, Ovol1 | Alx4, Ar, Ebf2, Esx1, Foxp2, Gm14393, Lhx2, Pou3f2, Rarb, Snai2, Sox8, Tead4, Tlx1, Zfhx4, Zfp52, Zfp532 |
| Grün et al. 2015 | Intesitinal organoid enteroendocrine cell | **Neurog3**, Fev, **Neurod1**, **Insm1**, **Nkx2-2**, **Foxa1**, Ets1, **Pax4**, **Lmx1a**, Pbx1, **Foxa2**, Hoxb2, Creb3l3, Msx1, Nfe2l2 | Evx1, Hsf5, Ikzf2, Irx3, Lef1, Obox3, Peg3, Sall2, Sp8, Zfp14, Zfp867 |
| Chu et al. 2016 | H9 ESC | **NANOG, POU5F1, SOX2** | DMRTB1, EGR4, HES3, INSM1, NKX2-6, OLIG3, PAX9, PITX3, SIX6, TFAP2B |
| Scialdone et al.[14] | Embryonic blood progenitor | Id3, Hes1, **Gata2**, Peg3, **Runx1**, **Fli1**, **Tal1**, **Gfi1b**, Klf6, **Sox7**, Ikzf1, Zfp367, Litaf, Gmeb1, Sr | Dbx2, Emx2, Gfi1, **Gfi1b**, Hsf3, Ikzf3, Myog, Nr5a1, Prdm13, Zfp541 |
| Scialdone et al.[14] | Embryonic visceral endoderm | Peg3, Ybx1, **Otx2**, **Eomes**, Arid1a, Arid3b, Foxq1, **Zic3**, **Foxa2**, **Hnf4a**, Zfp948, Hes1, Klf6, Hsf2, Elf2 | Creb3l3, En1, Foxa1, **Foxa2**, Foxa3, Foxq1, Gsc, **Hnf1b**, **Hnf4a**, Six3 |
| Gokce et al. 2016 | Striatal neuron | **Myt1l**, **Meis2**, **Bcl11b**, Peg3, Rarb, Rxrg, Aff4, **Foxp2**, Hivep2, **Bcl11a**, Arid4a, Six3, Dnajc2 | Ar, Atoh7, **Barhl1**, Cebpe, Foxa3, Gbx1, Glis1, Gm14139, Gn5294, Bhlha9 (and more) |
| Gokce et al. 2016 | Striatal microglia | **Egr1**, Fos, Zfhx3, Maf, Csde1, Mafb, Nfia, Junb, **Hmgb1**, Mef2c, **Sall1**, Atf4, Foxn3, Arid1a, Sub1 | Arid3c, Atoh1, Batf3, Ebf3, Hoxd10, Mlxipl, Pax7, Pitx2, Sox15, Zfp69 |
| Gokce et al. 2016 | Striatal vascular endothelial cell | Csde1, Hmgb1, Nfia, **Id3**, **Klf13**, Tcf4, Fos, **Klf6**, Jun, Gatad1, Tsc22d3, Fosb, **Klf4**, Arid1b, Zfp148 | Alx1, Erg, Foxc2, Foxl2, Nfatc4, Tbx1, Tbx2, Tbx4, Tcf21, Vsx1 |
| Joost et al. 2016 | Upper hair follicle I | Jun, **Klf4**, **Sox9** | Prdm14, Tbx19, Alx1, Insm2, Gm9376, Pou4f3, Prrx2, Rex2, Obox6, Sox3, Rnf138rt1 |
| Segerstolpe et al. 2016 | Pancreatic alpha cell | **MAFB**, NEUROD1, CNBP, TSC22D1, FEV, PAX6, IRX2, **ARX**, MLXIPL, CDIP1, PIAS1, HIF1A, ZNF655, TOX4, TULP4 | EVX1, FOXD4L3, GBX2, HOXC11, IFNB1, MEF2B, MYF6, POU3F3, SP9, SSX1 |
| Segerstolpe et al. 2016 | Pancreatic beta cell | ENO1, **CTNNB1**, **MAFA** | BHLHE23, DBX1, FERD3L, FOXR2, HOXB8, MYF5, OTP, SOHLH2, SOX3, TFAP2B |
| LaManno et al. 2016 | Fetal dopaminergic neuron type 2 | **NR4A2**, BNC2, TUB, **FOXA1**, POU6F1 | ALX4, ASCL2, ENF, FEZF1, FOXH1, NKX2-1, PRDM12, RAX2, TBX15, TBX22 |
| LaManno et al. 2016 | Fetal progenitor medial floorplate | **FOXA2**, **OTX2**, **LMX1A**, HMGA1 | FOXB1, FOXD4L1, GFI1, HNF1A, MESP2, NROB1, NR5A2, TBX5, ZNF99 |

neurons thought to give rise to *substantia nigra* DA neurons postnatally[4]. The synergistic transcriptional core of hDA2 neurons identified NR4A2, a nuclear receptor that controls mDA neuron identity and survival in mice[30], and FOXA1, a TF that together with FOXA2 contros mDA identity and neurogenesis in mice[31]. Finally, the hProgFPM synergistic core included TFs previously identified in the mouse midbrain floor plate and important for mDA neuron development in mice, such as FOXA2, OTX2, LMX1A[10, 32–34] which have not been previously recognized as the core of hProgFPM (Table 1; Supplementary Data 1). Overall, these examples demonstrate that synergistic transcriptional cores identified by TransSyn recapitulated known TFs controlling cell type/subpopulation identities along with their known functional, potentially synergistic interactions.

**Evaluation of TransSyn performance**. For an unbiased assessment of TransSyn performance, we calculated the percentage of cell subpopulations where at least one predicted TF has previously been experimentally validated to define the identity of that cell subpopulation. This showed that TransSyn could predict at least one such TF for 85 % of the cell subpopulations, for which at least one experimentally validated TF is known. We followed this criterion since the current knowledge of experimentally validated TFs is not complete (i.e., previously not tested) and includes TFs which are not classified as identity TFs according to our definition. The compiled list of TFs known to maintain cell subpopulation identities is shown in (Supplementary Data 2).

Importantly, we observed that pair-wise mutual information (MI) was not able to capture all the interactions among TFs in synergistic cores, supporting that these TFs interact synergistically rather than pair-wise (Supplementary Data 3). For example, this was observed in the case of interaction between the three plutipotency TFs (NANOG, POU5F1, and SOX2) in hESC, and Runx1, Fli1, Gata2, and Tal1 in the blood progenitor subpopulation described above, due to the multifactorial nature of the transcriptional regulatory mechanism. On the contrary, a set of TFs exhibiting pair-wise interactions among themselves does not necessarily display a multiple synergistic interaction, and therefore will not represent a synergistic transcriptional core. To show this, we performed a topological analysis of subpopulation specific GRNs inferred from pair-wise co-expression to identify top 10 subpopulation-specific hubs that could potentially be TFs that define subpopulation identities. Results showed that only a few known TFs were recovered as unique hubs (Table 2; Supplementary Data 4), indicating that transcriptional synergy is more suited for unraveling TFs that define subpopulation identities.

Next, we compared the performance of TransSyn to a method for identifying candidate identity TFs for bulk cell/tissue types using Jensen-Shannon Divergence (JSD)[7]. Since JSD was computed from bulk microarray data in this earlier study, we computed JSD using the average single-cell gene expression in each cell subpopulation. Results showed that in general, JSD predicted at least one TF in 33 % cell subpopulations in contrast to 85% achieved by TransSyn (Table 1A; Supplementary Data 5).

**Table 2 Unique top 10 hub TFs in GRNs for the example subpopulations in Table 1. Known identity TFs are in bold.**

| Data set | Cell subpopulation | Unique top 10 hub TFs |
|---|---|---|
| Treutlein et al. 2014 | Lung surfactant-secreting cuboidal alveolar type 2 cell | 2610008e11rik, Bcl11b, Crem, E2f3, Elf2, Foxq1, Gfi1, Hsf1, Ikzf2, Ikzf4 |
| Grün et al. 2015 | Intestinal organoid enterocyte precursor | Arx, Esrrg, Foxd2, Hoxa1, Hoxa4, Neurod2, Sox7, St18, Tox3, Zfp532 |
| Grün et al. 2015 | Intesitinal organoid enteroendocrine cell | 2700081O15rik, 5730507C01rik, Arntl, Atf6b, Dnajc2, Ehf, Erf, Etv5, Fiz1 |
| Chu et al. 2016 | H9 ESC | AEBP2, ARID1A, HIF1A, MIER1, TCF4, TSC22D2, ZNF146, ZNF286A, ZNF441, ZNF814 |
| Scialdone et al.[14] | Embryonic blood progenitor | Dnajc2, E2f3, **Tal1** |
| Scialdone et al.[14] | Embryonic visceral endoderm | Dmrta2, Elf4, Fosl1, Foxl1, Foxp1, Glis3, Klf14, Klf4, Smad9, Vdr |
| Gokce et al. 2016 | Striatal neuron | Csde1, Dbp, Erf, Gatad1, Hmgb1, Hsf2, Jund, Mier3, Thrb, Zfhx3 |
| Gokce et al. 2016 | Striatal microglia | Bhlha15, Lef1, Prox2, Tbx3, Zbtb17, Zfp113, Zfp184, Zfp82 |
| Gokce et al. 2016 | Striatal vascular endothelial cell | Arx, Sox21, Sp4, Tfap4, Tox3, Tshz2, Vsx1, Zfp433, Zfp579, Zfp709 |
| Joost et al. 2016 | Upper hair follicle I | Creb3l3, Lhx9, Rhox3f, Tal1, Zscan20 |
| Segerstolpe et al. 2016 | Pancreatic alpha cell | CARM1, DEAF1, JUNB |
| Segerstolpe et al. 2016 | Pancreatic beta cell | LDB1, **NKX6-1**, REPIN1, SREBF1 |
| LaManno et al. 2016 | Fetal dopaminergic neuron type 2 | AFF2, DACH2, FOXJ2, MAF, MEF2D, ZBTB48, ZNF354B, ZNF555, ZNF771 |
| LaManno et al. 2016 | Fetal progenitor medial floorplate | AHR, ARID3A, BARHL2, CEBPD, FOXF2, PRDM13, SCRT1, ZNF497, ZNF557 |

This result shows that TransSyn is more suited for identifying TFs that define closely related cell subpopulations. A systematic comparison with other tools, such as CellNet[5] and Mogrify[6], was not possible, since they do not currently consider user input single-cell RNA-seq data. Indeed, their built-in cell/tissue types exhibit a very limited overlap with the cell subpopulations we collected in this study. In particular, CellNet shares no overlap, while Mogrify shares a very limited overlap (Supplementary Table 1). The reprogramming factors identified by the latter for ESCs, NSCs, pancreatic mast cells and endothelial cells contained known identity TFs, whereas the factors for neurons from NSCs and between lung fibroblast and bronchial epithelial cells did not contain any known TF.

**Experimental validation of predicted identity TFs.** Finally, to demonstrate the usefulness of TransSyn, we carried out an experiment to shift the identity of hidbrain hNES cell line (SAI2)[35] to that of midbrain hProgFPM cells[4]. We first generated single-cell RNA-seq data of hNES cells, and found that its synergistic transcriptional core is quite different from that of hProgFPM cells (Fig. 2a). Analysis of the TFs required to convert hNES cells into hProgFPM cells identified OTX2, LMX1A, and FOXA2 (Fig. 2a). Since OTX2 is known to induce LMX1A[36], the conversion was performed by inducing the expression of the other two TFs, OTX2, and FOXA2. This was achieved by treating hNES cells during proliferation (FGF2+EGF) with two factors: (i) The small molecule smoothened agonist (SAG, 500 nM), which directly activates Shh signaling[37] and induces FOXA2[38]. (ii) The Wnt antagonist Dickoppf1 (Dkk1, 150 ng/ml), to reduce Wnt/β-catenin signaling to the levels required to induce OTX2[10] and midbrain dopamine neuron development[39] (Fig. 2b). Our expectation was that SAG would ventralize hNES cells and change their basolateral identity[35] into floor plate cells expressing FOXA2; and Dkk1 would anteriorize hindbrain cells expressing GBX2 into midbrain cells expressing OTX2[40]. Our results show that treatment of proliferating hNES cells with SAG and Dkk1 did not change the levels of the common midbrain-hindbrain TFs engrailed1 (EN1) and PAX2 (Fig. 2c, d), but increased the ratio of OTX2:GBX2 expression (Fig. 2e), indicating efficient anteriorization and acquisition of midbrain identity. In addition, we also observed increased levels of FOXA2 (Fig. 2f) and decreased levels of lateral genes, such as PAX6 and IRX3 (Fig. 2g, h),

indicative of efficient ventralization. These results were also confirmed by immunohistochemistry, which showed increased numbers of OTX2-positive cells (Fig. 2i).

To further confirm that the identity of the hNES cells had become that of hProgFPM cells, we tested their function, as assessed by their capacity to induce the expression of LMX1A at a later time-point and differentiate into midbrain DA neurons, reasoning that cells with the correct identity will be more efficient at generating DA neurons than the parental cells. Differentiation involved the removal of mitogens (FGF2 and EGF), as well as treatment with well-know midbrain patterning and differentiation factors such as Shh, Wnt5a, BDNF, GDNF, TGFβ3, and Wnt5a (reviewed in ref.[40]). In addition, we tested whether treatment with FGF8, a factor that was recently found to improve midbrain patterning and differentiation in human ES cells[41] was capable of further improving our protocol (Fig. 3a). We found that while both protocols strongly increased OTX2 and decreased GBX2 expression, only the protocol without FGF8 significantly increased LMX1A expression at day 8, as assessed by RT-qPCR (Fig. 3b). Similarly, both protocols increased the levels of NR4A2 and SLC6A3, but TH expression was only significantly increased by the SAG and Dkk1 protocol (Fig. 3c). Accordingly, while control unconverted cells were only capable of giving rise to rare and weak TH+ cells, cells differentiated after SAG and Dkk1 treatment gave rise to abundant TH+ neurons (Fig. 3d), and significant increased in the number of OTX2+ cells (Fig. 3e) and of TH+ cells (Fig. 3f). Moreover, TH+ cells were also LMX1A+, NR4A2+, and PBX1+ (Fig. 3g–i), confirming their midbrain identity[30, 33, 46]. In addition, TH+ cells expressed the mature neuronal marker MAP2 (Fig. 3j) and some of them were found to acquire a mature neuronal morphology with long processes and varicosities and bipolar morphology, typical of mDA neurons (Fig. 3k). Thus, our results show that by switching the identity of hNES to hProgFPM prior to differentiation, it is possible to rapidly differentiate hNES into DA neurons.

**Discussion**

In this study we postulated that cell subpopulation identity is determined by TFs that exhibit transcriptional synergy. Based on this proposition, we developed a computational method that dynamically searches for optimal synegistic transcriptional cores using an information theoretic measure of synergy computed

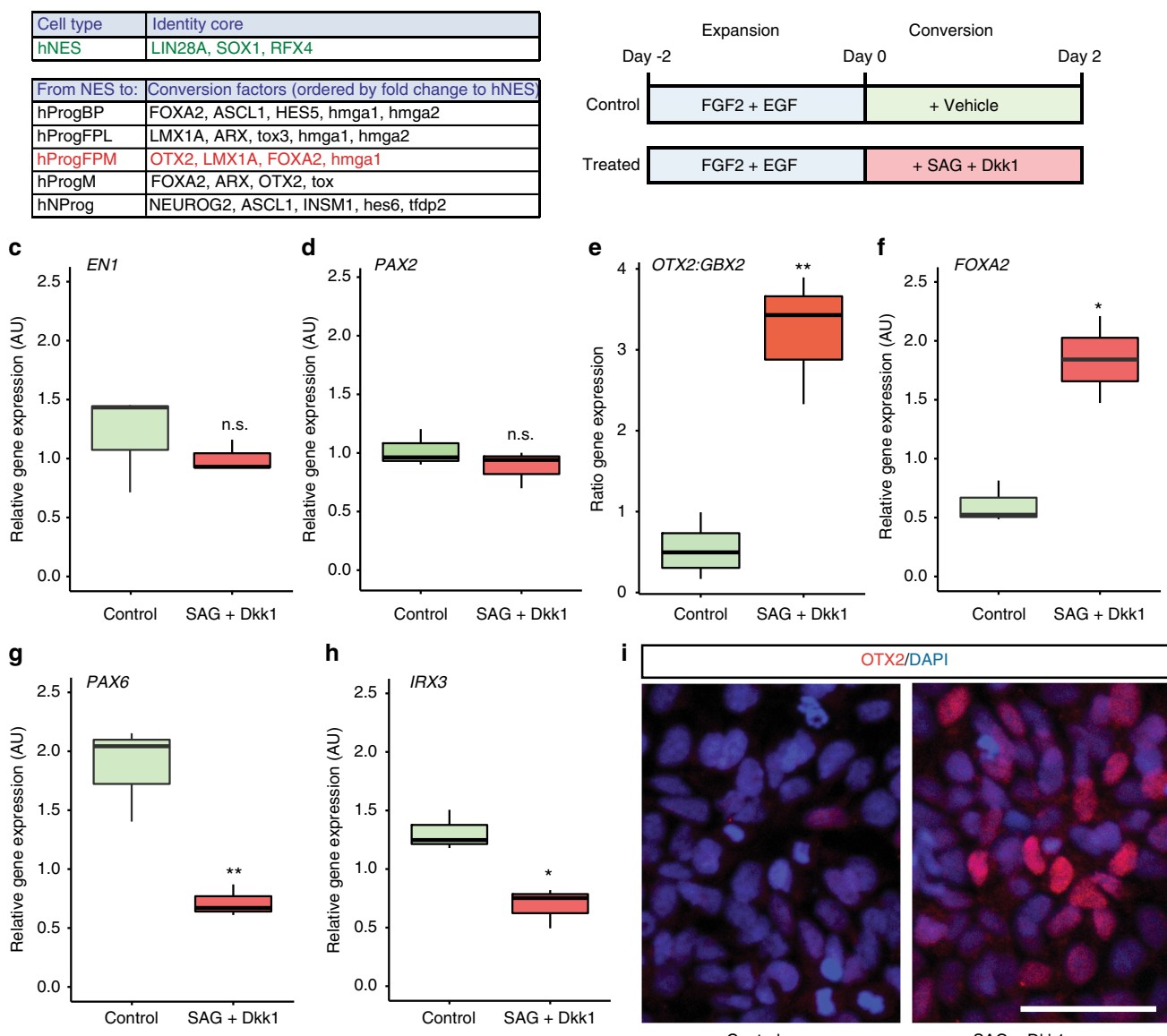

**Fig. 2** Conversion of basal hNES cells into medial floor plate midbrain progenitors (hProgFPM) by treatment of proliferating hNES for 2 days with the smoothen agonist (SAG, 500 nM) and Dickoppf1 (Dkk1, 150 ng/ml). **a** Transcription factors forming the synergistic transcriptional core of hNES cells and required for their conversion to midbrain progenitors. **b** Schematic representation of the treatment followed to convert proliferating hNES into HProgFPM, which included SAG, smoothen agonist (500 nM) and Dkk1, Dickoppf1 (150 ng/ml). **c–f** RT-qPCR analysis at day 2 showing the expression of TFs that define cell populations in the midbrain and hindbrain regions: *EN1* (**c**) and *PAX2* (**d**); the ventral midbrain: *OTX2:GBX2* ratio (**e**) and *FOXA2* (**f**), as well as more lateral compartments: *PAX6* (**g**) and *IRX3* (**h**). **i** Comparison of OTX2+cells in control and SAB+Dkk1-converted NES cells. Scale 50 μm. Box plots (**c–h**): Center line, median; hinges, 25% and 75% quartiles; whiskers, 1.5 interquartile range. Statistics: *t*-test; *$p \leq 0.05$; *$p \leq 0.01$. $N = 3$

from single-cell RNA-seq data. The predicted transcriptional cores recapitulatd known identity TFs in 85% of the tested cases and known synergistic TF interactions that relate to cell identity. Thus, the concept of transcriptional synergy employed in TransSyn represents a novel approach to specifically identifying transcriptional cores defining cell subpopulation identities. Following the experimental validation of the predicted identity transcriptional core of hProgFPM cells, we compiled a list of TFs whose up-/down-regulation may convert one cell subpopulation into another for 3786 pairs of initial and target cell subpopulations (Supplementary Data 6). Further validation of these transcriptional cores will reinforce the generality of the method. Importantly, unlike previously introduced methods, TransSyn does not require pre-compiled reference single-cell datasets,

which are unavailable for newly identified cell subpopulations. In addition, TransSyn does not rely on GRN inference and analysis, which could be a bottleneck for accurate predictions of identity transcriptional cores. In summary, such unbiased identification of synergistic transcriptional cores may facilitate the development of general strategies for cell subpopulation conversions, opening up novel functional applications in regenerative medicine, such as the generation of DA neurons for Parkinson's disease.

## Methods

**Single-cell RNA-seq data**. Single-cell RNA-seq data used in this study were obtained for the following biological systems; the mouse datasets for lung, striatum, cortex, and hippocampus, quiescent, and active NSCs, intestine, circulating pancreatic tumor cells, hair follicles, and gastrulating embryo, and the human datasets

for midbrain, CD127+ lymphoid cells, pancreas, ovarian cancer, germline cells, and in vitro hESCs. The reference to each dataset is described in (Supplementary Data 1). We used the same subpopulation classifications defined in the respective original studies. We also analyzed other datasets not listed here, however, they did not have an enough number of expressed TFs in the majority of cell subpopulations, and were therefore discarded. In addition, synergistic transcriptional cores for cell subpopulations that were either "undefined" or with less than three cells

were not considered. We did not reprocess each raw data and same gene expression values that were used in the original studies were also used in this study. TFs were considered "expressed" if their expression values were ≥1 in RNA-seq FPKM/RPKM/TPM values, ≥10 in normalized read counts, or ≥1 in UMI counts. TFs below these thresholds were considered "not expressed". Exceptionally, the expression cutoff of 10 was used for the hESC dataset, since setting it to 1 resulted in too many expressed TFs and the subsequent computation became infeasible.

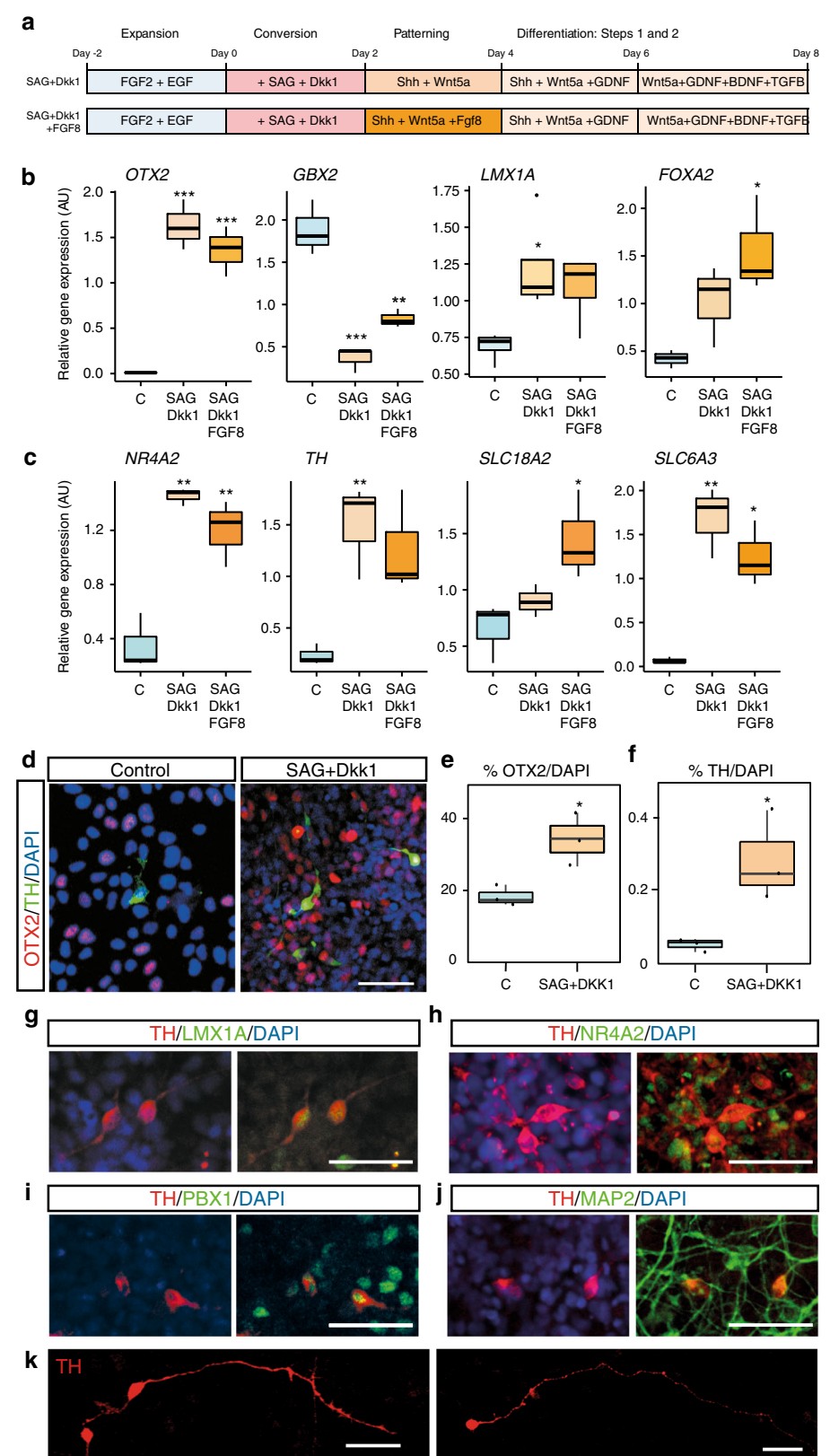

**Identification of most frequently expressed TFs**. The definition of TFs was obtained from the AnimalTF database[42]. The fraction of cells expressing each TF was computed in each subpopulation and the top 10% most frequently expressed TFs were shortlisted for further analyses. Among these TFs, we discarded those that were not expressed in more than 30% of cells. For the La Manno et al., 2016, dataset[4], the binarized expression status estimated in the original study was used. Since this filtering retained many TFs that were expressed at very low intensity, TFs with mean UMI count <1 were further discarded. Since the subsequent computation becomes infeasible for standard desktop computers if the number of TFs is more than 150, in these cases TFs with highest coefficient of variation were discarded to make the number of TFs ≤150.

**MMI computation**. In each cell subpopulation MMI[11] among the most frequently expressed TFs was computed by:

$$\mathrm{MMI}(S) = -\sum_{T \subseteq S} (-1)^{|T|} H(T),$$

where $S = \{X_1, X_2, \ldots X_n\}$, $T$ is a subset of $S$, and $|T|$ indicates the number of variables in this subset. In the current study these variables are discretized gene expression values of TFs. This equation becomes MI if only two variables are considered. In the case of three variables the equation can be written as:

$$\mathrm{MMI}(X; Y; Z) = H(X) + H(Y) + H(Z) - H(X, Y) - H(Y, Z) \\ - H(Z, X) + H(X, Y, Z)$$

To compute Shannon's entropy, gene expression values were first log10-transformed. Zero gene expression values were converted into 1 prior to the transformation. This value was then discretized within each cell subpopulation using the Freedman–Diaconis rule implemented in the R nclass.FD function and Shannon's entropy of each TF was computed on these discretized values. The input value for the nclass.FD function was set to the number of cells +1 for FPKM/RPKM/TPM values, and normalized read counts, while the number of cells +6 was used for UMI counts. The range of gene expression value was set between 0 and maximum value of a given cell subpopulation. Since the bin size for each TF is different, the entropy was normalized by the theoretical maximum entropy (i.e., entropy when all bins contain an equal number of variable) to enable a direct comparison between different TF entropies. The MMI was then computed using all cells in the entire population of a given dataset except for the ones in the subpopulation for which MMI is being computed. As described above, the joint entropy was also normalized by the theoretical maximum entropy prior to MMI computation.

**Dynamic search for synergistic transcriptional cores**. MMI was first computed for all combinations of three TFs and the top one percent lowest MMI (i.e., synergistic) combinations were taken. Then, these TF combinations were ranked by TC defined as:

$$\mathrm{TC}(S) = \left( \sum_{X_i \subseteq S} H(X_i) \right) - H(S),$$

where $S = \{X_1, X_2, \ldots X_n\}$. TC measures the interaction strengths (MI) shared among all subsets of the variables within a combination, and is more appropriate for comparing interaction strengths between different combinations than MMI[12], which measures the information gain from the previous seed combination. Then, top one percent highest TC combinations were used as initial seeds for the subsequent search for higher-level synergistic combinations of TFs. To this end, new TFs were added to each seed combination one by one and MMI for the new combination was computed. Then, the combinations that showed lower (less than 0.05) MMI than the seed were taken to the next iteration. For example, if {A, B, C, D, E, F, G, H} were the selected TFs in a given cell subpopulation and {A, B, C} was a seed combination, then MMI of all 4 TF combinations, {A, B, C, D}, {A, B, C, E}, {A, B, C, F}, {A, B, C, G}, and {A, B, C, H}, were computed and if the difference in MMI between the new combination and seed was negative, then that new combination was kept. Then, these new, more synergistic TF combinations were again ranked by TC and the top 10 combinations were used as seeds for the identification of best 5 TF combinations next, and so on. This procedure was continued until no

new combination is more synergistic than the seed. We also terminated the procedure when the number of TFs reached 15, since continuing with more than this number was often computationally impractical. We think this operation is acceptable, since usually at this point most TFs are shared among different combinations. Once the search is terminated, MMI for all combinations of the top 20 best TC combinations is computed and if there is more synergistic combination(s), then those combinations are ranked by TC as the final synergistic transcriptional cores. If more than one top combination (i.e., ties) is present, they are ranked by the highest summed mean gene expression and the top three combinations were kept as the final synergistic transcriptional cores. For the identification of cell conversion TFs, TFs in the synergistic transcriptional core of a target cell subpopulation were ranked by the mean gene expression fold change between the target cell subpopulation and starting cell subpopulation. The main part of TransSyn was written in C++, which was wrapped in R using the Rcpp package.

**MI computation between TFs**. Pair-wise MI was computed for TF pairs in transcriptional synergistic cores, in which at least two TF are known to maintain that cell subpopulation. The gene expression values were first log2-transformed and then discretized within each cell subpopulation using the Freedman–Diaconis rule, as described above. Then Shannon's entropy of each TF and joint entropies of each pair of TFs were computed on these discretized values. MI was then computed by:

$$\mathrm{MI}(X; Y) = H(X) + H(Y) - H(X, Y),$$

The statistical significance of each edge was computed by a $t$-test against a null distribution formed by randomizing data 50 times and edges with the top 1% lowest $p$-value were kept as the final edges.

**GRN hub analysis**. A GRN for each cell subpopulation was inferred using the corresponding cell subpopulation single-cell RNA-seq data with four different algorithms, PCC, SCC, MRNET[43], and random forest-based method (GENIE3[44]). The default parameters were used for GENIE3. For RNA-seq FPKM/RPKM/TPM values and RNA-seq normalized read counts, the values were log2-transformed prior to the inference. No transformation was applied to UMI counts.

**JSD computation for TFs**. For each TF, JSD was computed between an ideal gene expression vector and an observed gene expression vector, as was previously performed in[7]. The ideal gene expression vector was formed by putting 1 to the query cell subpopulation and 0 to all other subpopulations within a dataset. The observed gene expression vector was formed by computing the average gene expression for each subpopulation and normalizing each value by the sum of the average gene expression values of all the subpopulations. The top 10 TFs were taken as the predicted identity TFs.

**Cultivation of Lt-NES SAI2 cells**. In our study we used the Long-term self-renewing neuroepithelial-like stem cells (Lt-NES) SAI2 line generated from human hindbrain fetal tissue[35]. Mycoplasma-free cells have been kept in proliferation according to previously described protocols[45], in 6-well plates coated with poly-L-ornithine (1:5 in water; Sigma) and laminin (1:500 in water, Invitrogen), using maintenance media based on DMEM F12 Glutamax Medium (GIBCO, Life-Technologies) supplemented with N2 (1:100, GIBCO, LifeTechnologies), B27 (1:1000, GIBCO, LifeTechnologies), and the growth factors hEGF (10 ng/ml, R&D) and FGF2 (10 ng/ml, R&D). To modify the identity of Lt-NES, cells were treated for 48 h with SAG (500 nM, Tocris) and Dkk1 (150 ng/ml, R&D) in the proliferation media. Treated and non-treated cells were compared.

For differentiation experiments, Lt-NES cells treated as above for 48 h were seeded at a density of 100.000 cells in 48-well plates coated with PLO and laminin. Cells were differentiated for 6 days in following the protocol described in ref.[46] with some modifications: Cells were patterned for 2 days in media containing N2 Supplement (1:100), B27 (1:1,000), Shh (200 ng/ml, R&D) and Wnt5a (100 ng/ml) with or without FGF8B (100 ng/ml, PeproTech). Cells were subsequently differentiated for 4 days on media containing N2 (1:100) and B27 (1:100). During the first 2 days in GDNF (20 ng/ml, R&D) and BDNF (20 ng/ml, R&D) and the last 2 days in GDNF (20 ng/ml, R&D), BDNF (20 ng/ml, R&D), dcAMP (0,5 mM, Sigma), Ascorbic Acid (200 μM, Sigma), and TGFβ3 (2 ng/ml, R&D).

**Fig. 3** Conversion of hNES cells into hPRogFPM and their differentiation into midbrain dopaminergic neurons. **a** Schematic representation of the conversion and differentiation protocols. **b, c** RT-qPCR analysis at day 8, showing the expression of midbrain-hindbrain TFs, such as *OTX2*, *GBX2*, *LMX1A*, and *FOXA2* (**b**), as well as the dopaminergic neuron markers, *NR4A2*, *TH*, *SLC18A22*, and *SLC6A3* (**c**). **d** Immunocytochemistry analysis of the presence of OTX2 and TH in control unconverted NES cultures, compared with NES cells converted with SAF+Dkk1 and differentiated until day 8. **e, f** Percentage of OTX2+ and TH+ cells in the conditions in **d**. $P = 0.02673$ (**e**), $P = 0.03233$ (**f**), $n = 3$. **g–i** Expression of the key midbrain TFs, LMX1A, NR4A2, and PBX1, in TH+cells derived from SAI2-NES cells after conversion and differentiation. **j, k** TH+ cells express the mature neuronal marker, MAP2 (**j**), and some acquire mature neuronal morphologies, with long processes and varicosities at day 8 (**k**). Scale 50μm. Box plots (**b, c, e, f**): Center line, median; hinges, 25% and 75% quartiles; whiskers, 1.5 interquartile range. Statistics: (**b, c**) ANOVA, followed by pair-wise t-test with Bonferroni correction for multiple testing. (**e, f**). Two sample t-test; *$P \leq 0,05$; **$P \leq 0,01$; ***$P \leq 0,001$. $N = 3$ (GBX2, FOXA2, TH, SLC6A3), $n = 4$ (LMX1A, OTX2, NR4A2, SLC18A2)

**hNES single-cell RNA-seq data**. Single cell RNA-seq of undifferentiated Lt-NES was obtained and analyzed from GSE114670.

**RT-qPCR**. RNA was extracted form Lt-NES SAI2 cells using RNeASY mRNA isolation system (Qiagen) according to the manufacturer's instructions and treated with DNAse on-column protocol. 200–500 ng of total RNA were reverse-transcribed using the Superscript II kit (Invitrogen). The reverse-transcribed cDNA was amplified using Fast SYBRGREEN (Applied Biosystems) in a StepONE plus real time-qPCR machine (Applied Biosystems). Outliers were detected by using the absolute deviation from the median, statistical significance was measured with the Welch two sample two tailed $t$-test and Bonferroni correction in case of multiple testing. Stars indicate *$P \leq 0,05$; **$P \leq 0,01$; ***$P \leq 0,001$.
Primers used in our analysis:
SLC6A3 Forward 5′---3′ ACCTTCCTCCTGTCCCTGTT
Reverse 3′---5′ CACCATAGAACCAGGCCACT
EN1 Forward 5′---3′ CGTGGCTTACTCCCCATTTA
Reverse 3′---5′ TCTCGCTGTCTCTCCCTCTC
FOXA2 Forward 5′---3′ TTCAGGCCCGGCTAACTCT
Reverse 3′---5′ AGTCTCGACCCCCACTTGCT
GAPDH Forward 5′---3′ TTGAGGTCAATGAAGGGGTC
Reverse 3′---5′ GAAGGTGAAGGTCGGAGTCA
GBX2 Forward 5′---3′ GTTCCCGCCGTCGCTGATGAT
Reverse 3′---5′ GCCGGTGTAGACGAAATGGCCG
IRX3 Forward 5′---3′ CTGACGAGGAGGGAAACGCTTA
Reverse 3′---5′ GAGCTCCTCCTCCTCCAGCTCT
LMX1A Forward 5′---3′ GATCCCTTCCGACAGGGTCTC
Reverse 3′---5′ GGTTTCCCACTCTGGACTGC
NR4A2 Forward 5′---3′ AGTCTGATCAGTGCCCTC
Reverse 3′---5′ CCCCATTGCAAAAGATGAGT
OTX2 Forward 5′---3′ ACAAGTGGCCAATTCACTCC
Reverse 3′---5′ GAGGTGGACAAGGGATCTGA
PAX2 Forward 5′---3′ TAGACTGCGGACTGGGGTCTTC
Reverse 3′---5′ GGTTCTTACCACCGGCAGATTG
PAX6 Forward 5′---3′ TGGTATTCTCTCCCCCTCCT
Reverse 3′---5′ TAAGGATGTTGAACGGGCAG
TH Forward 5′---3′ ACTGGTTCACGGTGGAGTTC
Reverse 3′---5′ TCTCAGGCTCCTCAGACAGG
SLC18A2 Forward 5′---3′ CACTGCCTCCATCTCAGACA
Reverse 3′---5′ CCGGTGACCATAGTCGAGTT

**Immunocytochemistry on differentiated hNES cells**. For immunocytochemical analysis, cells were fixed for 20 min at room temperature in 4% paraformaldehyde (PFA) in PBS, permeabilized and blocked for 60 min in PBS containing 0.3% Triton X-100, 0.1% BSA and 10% normal donkey serum (PBTA-NDS). Then they were incubated overnight at 4 °C in PBTA-NDS with different antibodies: rabbit TH (1:1000, Pel Freeze, P4010-0), mouse TH (1:250, Immunostar, 22941) or sheep TH (1:250, Novus, NB300-110), mouse MAP2 (1:100, Sigma, M4403), mouse PBX1 (1:200, Santa Cruz, SC-101851), rabbit NR4A2 (1:200, Santa Cruz, SC-990), rabbit LMX1A (1:4000, Millipore, AB10533), and goat OTX2 (1:1000, Bio-techne, AF1979). Next day the cells were washed three times with PBS and incubated for 2 h at room temperature with Alexa Fluor secondary antibodies (1:500, Invitrogen) 647 (A31571), 555 (A31572, A21432, A21436, 488 (A11035, A21467, A21206, A11015), and 4′,6-Diamidino-2-phenylindole dihydrochloride (DAPI, Sigma, D8417) in PBTA-NDS. Microphotographs were taken with a Zeiss LSM800 confocal microscope (CLICK facility, Karolinska Institute) using the same settings. Cell counts were performed in a blinded fashion in 3 independent experiments, and 6-9 randomly selected fields/condition. Control and experimental images were processed linearly, in the same way, using Fiji software (ImageJ version 1.51t) and Photoshop CS5 (Adobe System Inc.).

**Code availability**. TransSyn is freely available at https://sourceforge.net/projects/transsyn/.

**Data availability**. The single-cell RNA seq data of undifferentiated Lt-NES is available at GEO: GSE114670. The rest of the data supporting the conclusions of this study are available from the correspoing author upon request.

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

## Acknowledgements

S.O. is supported by an FNR CORE grant (C15/BM/10397420), S.R. by University of Luxembourg IRP Grant (R-AGR-3227-11), E.M.T. by a fellowship from the Swedish Research Council. This project was supported by the Swedish Research Council (VR 2016-01526), Swedish Foundation for Strategic Research (SRL program and SB16-0065), European Commission (NeuroStemcellRepair), Karolinska Institutet, Cancerfonden (CAN 2016/572), and Hjärnfonden (FO2015:0202). We would also like to thank Gioele La Manno and Peter Lönneberg for help with single cell data; and the Knut and Alice Wallenberg Foundation for support to the CLICK imaging facility at KI.

## Author contributions

A.d.S. conceived the overall study and designed its computational part, S.O. and S.R. developed the computational method, compiled data sets and performed the analysis, E. A. designed the cell culture experiments. C.S., S.Y., and E.M.T. performed the experiments, S.O., C.S., S.R., E.A., and A.d.S. wrote the manuscript.

## Additional information

**Competing interests:** The authors declare no competing interests.

