## [Peer Review File · Nature Communications]

Reviewers' comments:

Reviewer #1 (Remarks to the Author):

Reviewer's comments for Okawa et al., "Transcriptional synergy as an emergent property defining cell subpopulation identity and its application to population shift".

Here, the authors present a computational platform designed to identify the synergistic transcriptional cores that determine cell identity. Via the search for optimal synergistic transcriptional cores, the authors predict transcription factor combinations to drive conversion of cell identity. They apply this approach with an aim to improve the efficiency of conversion of human hindbrain neuroepithelial cells into medial floor plate midbrain progenitors. Overall, TransSyn represents an elegant computational approach to identify key transcription factors to direct changes in cell identity. Although experimental validation is included in the manuscript, it would be more convincing to see the improved efficiency of conversion driven by the factors alone, rather than indirectly via manipulation of cell signaling pathways.

Specific comments:

- In the introduction, it would be helpful to explicitly state the utility of the platform for understanding changes in cell identity and how it can be artificially manipulated.
- TransSyn considers all possible direct and indirect regulatory interactions that can be measured by gene expression. This is conceptually very elegant, but how does this work so precisely with technically noisy data such as scRNA-seq?
- The examples provided, where TransSyn finds POU5F1, NANOG and SOX2 as the most synergistic transcriptional core in hESCs, and Tal1, Gata2 and Hhex in blood progenitors is a very nice validation. To expand on this, are there examples of reprogramming strategies where the factors used for cell fate conversion overlap with synergistic transcriptional cores revealed by TransSyn?
- In table S6, it would be helpful to expand on the cell names.
- In terms of the experimental validation provided, the images supplied are not sufficient to demonstrate an increase in efficiency. A more quantitative assay should be performed.
- The choice of experimental validation seems tangential. Why not attempt to boil an increased efficiency down to the TransSyn-identified factors? Manipulation of signaling can target many genes, leading to indirect effects. It would be a much more powerful demonstration to only manipulate the target transcription factors.

Reviewer #2 (Remarks to the Author):

The manuscript describes a method for identification of synergistic TFs in cell sub-populations. Since the authors assert that this is the first publication of such a method (with the novelty focusing on the application to single-cell sequence data and to cell sub-populations) the background/introduction section should cite previous related work (such as 32,33) that focuses on cell type. As such a clear explanation to the reader of the definition of a sub-population versus a cell type, a description of why the existing cell-type methods cannot be applied to sub-populations and/or single-cell sequence data is needed. Thus the case for the novelty of this work will stand out from the beginning, and the context within the field will be clear.

The authors do have one sentence in the discussion addressing this, but the argument is weak and inaccurate; not all these methods require GO, and GRNs used are not sub-population specific. In fact a completely different argument is given against GRNs in the introduction (with no reference to existing methods). Given the present arguments against existing methods, as a reader I cannot see why they cannot be applied to the same data and benchmarks for comparison to TransSyn.

For those methods for which a comparison is possible, it must be carried out as it has been for JSD, and for those where it is not possible (32?,33?), a more complete explanation must be given than that provided. The current explanation is flawed and it is not stated that comparison (e.g. to CellNet) is not possible. For a methods paper, this is essential.

Apart from this isolation from previous work and lack of systematic comparison, the paper and the work is excellent, with very exciting experimental validation. Single-cell sequencing is certainly a topic of high general interest at the moment, and this paper could be well received.

On the related topic of distinguishing this paper from other work, it would be interesting to hear more about differences using single-cell data, and important to understand what is fundamentally and intrinsically different about sub-populations of cells versus different cell types. The authors present the differences as fundamental, but it is not explained why. The authors claim a 'conceptual advancement' without discussion of how their concept is different from cell-type work that precedes it; it does look like application of an existing concept to single-cell data.

Reviewer #3 (Remarks to the Author):

Okawa et al. developed a computational platform (TransSyn) for identifying synergy of transcription factors in single cell RNA-seq data. The advantage of their platform is that it only requires a large set of single-cell RNA-seq data from distinct subpopulations as input and does not require pre-compiled reference datasets. The TransSyn can find some critical synergistic transcription factors, like NANOG, POU5F1, SOX2 for ESCs and Myt1l for neurons.

The authors demonstrated the superiority of their system in predicting the combination of TFs for cell identity when compared with JSD method. They should also compare with a newly developed method developed by Rackham et al (Nature Genetics, 2016) even though that method was developed to predict core TFs for transdifferentiation.

The example for the application of the synergistic transcription factors is not convincing. There is no data showing that FGF-EGF expanded progenitors hNES exhibit the hindbrain identity. So, the rationale behind the use of DKK1 to switch to the midbrain fate is not clear. No quantification of TH+ neurons and other DA neuron markers, such as NR4A2 and PITX3v are provided to support their claim of "more efficient differentiation towards dopaminergic neurons than standard protocols". Additional proof by directing the cell fate by transgenic expression of transcription factors OTX2, LMX1A, FOXA2 is a plus.

We thank all the reviewers for constructive comments. Our manuscript has significantly benefited from them. Please see below our response to each reviewers comment in blue. All major changes in the revised manuscript are highlighted in yellow.

Reviewers' comments:

Reviewer #1 (Remarks to the Author):

Reviewer's comments for Okawa et al., "Transcriptional synergy as an emergent property defining cell subpopulation identity and its application to population shift".

Here, the authors present a computational platform designed to identify the synergistic transcriptional cores that determine cell identity. Via the search for optimal synergistic transcriptional cores, the authors predict transcription factor combinations to drive conversion of cell identity. They apply this approach with an aim to improve the efficiency of conversion of human hindbrain neuroepithelial cells into medial floor plate midbrain progenitors. Overall, TransSyn represents an elegant computational approach to identify key transcription factors to direct changes in cell identity. Although experimental validation is included in the manuscript, it would be more convincing to see the improved efficiency of conversion driven by the factors alone, rather than indirectly via manipulation of cell signaling pathways.

Specific comments:

- In the introduction, it would be helpful to explicitly state the utility of the platform for understanding changes in cell identity and how it can be artificially manipulated.

Identification of subpopulation specific transcriptional cores can facilitate designing strategies for experimental conversion of cell subpopulation identities. This was actually stated in the 2nd and 3rd paragraphs. Nevertheless, following this advice we have more explicitly stated the utility of the proposed approach in the introduction of the revised manuscript.

- TransSyn considers all possible direct and indirect regulatory interactions that can be measured by gene expression. This is conceptually very elegant, but how does this work so precisely with technically noisy data such as scRNA-seq?

Several existing methods attempt to model and circumvent technical noise in single-cell sequencing data. Further, data quality also depends on the technique used to acquire and process the samples. In this regard, specifically addressing technical noise is not in the scope of TransSyn, as it just requires normalized single-cell RNA-seq data as input. However, it initially filters away genes that are not expressed in more than the minimum required number of cells in each subpopulation. This step can make our approach less susceptible to technical noise in the form of dropout events. Another possible reason is

that calculation of synergy based on multivariate mutual information can reduce the effect of noise, since an increase in dimensionality will progressively reduce the uncertainty, including technical noise (Srinivasa, S. (2005). A review on multivariate mutual information (Tech. Rep. No. EE-80653), University of Notre Dame). These two aspects of TransSyn likely help alleviate the technical noise problem.

- The examples provided, where TransSyn finds POU5F1, NANOG and SOX2 as the most synergistic transcriptional core in hESCs, and Tal1, Gata2 and Hhex in blood progenitors is a very nice validation. To expand on this, are there examples of reprogramming strategies where the factors used for cell fate conversion, overlap with synergistic transcriptional cores revealed by TransSyn?

In the case of human oculomotor and trochlear nucleus neurons (hOMTN), TransSyn identified ISL1 and PHOX2A as part of the core. These two TFs are known to synergistically specify cranial motor neurons from mESCs (Mazzoni et al. 2013). We added this example to the result section of the revised manuscript. Apart from this, we could not find any other case in our datasets where the synergistic cores contain a set of known reprogramming factors.

- In table S6, it would be helpful to expand on the cell names.

This is a good point. Now we have expanded on the cell names. The other relevant tables were also expanded accordingly. Arbitrary IDs and numbers used in the original studies were kept, so that any interested reader can go back and identify exact subpopulations in the original papers.

- In terms of the experimental validation provided, the images supplied are not sufficient to demonstrate an increase in efficiency. A more quantitative assay should be performed.

Following the suggestion by this reviewer, we have performed a detailed and quantitative analysis of the phenotype of the differentiated cells (new Figure 3). In this section we used a protocol in which NES cells are treated for 2 days with Dkk1 and SAG (conversion) and differentiated for 6 days (Fig 3A). To determine whether midbrain patterning with Dkk1 and SAG is sufficient or could be further enhanced, we compared the standard protocol (Dkk1 and SAG) with and without the addition of FGF8, a factor that has been recently reported to improve midbrain patterning and differentiation of human ES cells (Kirkeby et al., 2017). Our results show that while both protocols further increased OTX2 and decreased GBX2 expression (compared to 2 days, Fig 2), the addition of FGF8 prevented a sufficient increase of LMX1A expression, a TF critical for midbrain dopaminergic differentiation (Andersson et al., 2006). In addition while both protocols increased midbrain dopamine markers such as NR4A2 and DAT, only SAG and Dkk1 treatment (in the absence of FGF8) significantly increased (by 5-fold) the number of TH+ cells. These results indicate that FGF8 is not required to increase the overall efficiency of our protocol and suggested that Dkk1 and SAG are sufficient to change the identity of NES cells.

To further confirm these results, we performed immunocytochemistry at day 8 of differentiation and found rare, immature and very weak TH+ cells in the control condition and a strong increase in the number and morphological complexity of TH+ cells after SAG and Dkk1 treatment (Fig 3D,E). This increase was accompanied by an increase in the number of OTX2+ cells (Fig 3D,F). Moreover, we also found that TH+ cells in the SAG+ Dkk1 condition co-express the midbrain markers NURR1 and LMX1A. Combined, our results of the analysis of TF expression and functionality after conversion and differentiation of NES cells indicate that SAG and Dkk1 treatment induces the expression of the TFs that define the subpopulation identity identified by TransSyn and, as predicted, convert NES cells into medial midbrain floor-plate progenitors, capable of generating dopamine neurons.

- The choice of experimental validation seems tangential. Why not attempt to boil an increased efficiency down to the TransSyn-identified factors? Manipulation of signaling can target many genes, leading to

indirect effects. It would be a much more powerful demonstration to only manipulate the target transcription factors.

We thank the reviewer for the comment. In the present study we decided not to use forced expression of the identity TFs (i.e reprogramming) because this strategy would limit our capacity to use these TFs to evaluate the acquisition of the correct cell identity. Instead, we chose to use morphogens and small molecules to change the identity of the NES cells and induce their differentiation. We think this is a better option, as the acquisition of the correct cell identity can be simply done by analyzing the expression of endogenous identity TFs. Another reason to follow this approach is that morphogens and small molecules are the current tools of choice for the development of stem cell differentiation protocols aiming at generating midbrain dopaminergic neurons for transplantation in PD. In addition, the manipulations performed in this study directly activate the signaling pathways controlling cell identity. SAG (Smoothed agonist) is known to activate the Shh receptor smoothed (Chen et al. 2002) and induce FOXA2 (Denham et al., 2012), which is required for floor plate development (Sasaki and Hogan 1994). On the other hand, Dkk1 directly modulates Wnt signaling (Glinka et al., 1998) to reach the levels required to induce OTX2 (Chung, et al. 2009) and midbrain dopamine neuron development (Ribeiro et al., 2011). Our data also shows that treatment of hNES cells with SAG+ Dkk1 results in the expected increases in FOXA2:GBX2 ratio and OTX2 (Figure 2), supporting the notion that the compounds we used target the TFs predicted by TransSyn to define hProgFPM identity. This aspect is now more clearly explained in the results section.

References:

- Srinivasa, S. A review on multivariate mutual information. 2005 (Tech. Rep. No. EE-80653). Notre Dame, IN: University of Notre Dame.
- Mazzoni EO,, Mahony S, Closser M, Morrison CA, Nedelec S, Williams DJ, An D, Gifford DK, Wichterle H. Synergistic binding of transcription factors to cell-specific enhancers programs motor neuron identity. *16*, 1219–1227 (2013). doi:10.1038/nn.3467 pmid:23872598
- Andersson E, Tryggvason U, Deng Q, Friling S, Alekseenko Z, Robert B, Perlmann T, Ericson J. Identification of intrinsic determinants of midbrain dopamine neurons. *Cell*. 2006 Jan 27;124(2):393-405
- Chen JK, Taipale J, Young KE, Maiti T, Beachy PA. Small molecule modulation of Smoothed activity. *Proc. Natl Acad. Sci. USA* 2002 **99**, 14071–14076
- Chung S, Leung A, Han BS, Chang MY, Moon JI, Kim CH, Hong S, Pruszk J, Isacson O, Kim KS. (2009). Wnt1-lmx1a forms a novel autoregulatory loop and controls midbrain dopaminergic differentiation synergistically with the SHH-FoxA2 pathway. *Cell Stem Cell* 5, 646-658.
- Denham M, Bye C, Leung J, Conley BJ, Thompson LH, Dottori M. (2012) Glycogen synthase kinase 3 β and activin/nodal inhibition in human embryonic stem cells induces a pre-neuroepithelial state that is required for specification to a floor plate cell lineage. *Stem Cells* 30(11):2400-11.
- Glinka A, Wu W, Delius H, Monaghan AP, Blumenstock C, Niehrs C. (1998) Dickkopf-1 is a member of a new family of secreted proteins and functions in head induction. *Nature* 391(6665):357-62.
- Kirkeby A, Nolbrant S, Tiklova K, Heuer A, Kee N, Cardoso T, Ottosson DR, Lelos MJ, Rifles P, Dunnett SB, Grealish S, Perlmann T, Parmar M. Predictive Markers Guide Differentiation to Improve Graft Outcome in Clinical Translation of hESC-Based Therapy for Parkinson's Disease. *Cell Stem Cell*. 2017 Jan 5;20(1):135-148.
- Ribeiro D, Ellwanger K, Glasgow D, Theofilopoulos S, Corsini NS, Martin-Villalba A, Niehrs C, Arenas E. (2011) Dkk1 regulates ventral midbrain dopaminergic differentiation and morphogenesis. *PLoS One*. 6(2):e15786.

Sasaki H, Hogan BL. (1994) HNF-3 beta as a regulator of floor plate development. Cell 76(1):103-15.

Reviewer #2 (Remarks to the Author):

The manuscript describes a method for identification of synergistic TFs in cell sub-populations. Since the authors assert that this is the first publication of such a method (with the novelty focusing on the application to single-cell sequence data and to cell sub-populations) the background/introduction section should cite previous related work (such as 32,33) that focuses on cell type. As such a clear explanation to the reader of the definition of a sub-population versus a cell type, a description of why the existing cell-type methods cannot be applied to sub-populations and/or single-cell sequence data is needed. Thus the case for the novelty of this work will stand out from the beginning, and the context within the field will be clear.

We agree that this point was not clearly mentioned in the original manuscript. The important point here is that the definition of identity TFs depends on the context. For example, in the context of cell/tissue types such as neurons and hepatocytes, the definition of identity TFs comes down to the comparison between these largely different cell types. However, in the context of cell subpopulations within a tissue or organ, such as different subtypes of neurons, the definition of identity TFs becomes subtler due to the increased commonality between them. Existing methods for identifying cell fate determinants (e.g. ref. 32 and 33) rely on a set of pre-compiled gene expression profiles of bulk cell/tissue types. Consequently, the application of these methods is limited to those pre-compiled bulk cell/tissue types, and cannot be applied to novel subpopulations of cells identified in a newly generated single-cell dataset. For example, these methods could not predict cell conversion factors from hNES cells to hProgFPM cells, as we successfully did in this study.

We have provided a more description of cell subpopulation identity, its distinction from cell/tissue types and the previous existing methods in the introduction of the revised manuscript.

The authors do have one sentence in the discussion addressing this, but the argument is weak and inaccurate; not all these methods require GO, and GRNs used are not sub-population specific. In fact a completely different argument is given against GRNs in the introduction (with no reference to existing methods). Given the present arguments against existing methods, as a reader I cannot see why they cannot be applied to the same data and benchmarks for comparison to TransSyn.

The reviewer is right in that ref. 32 (CellNet) does not require GO. Ref. 33 (Mogrify) needs it but not the standard GO but their own version (FANTOM5 cell ontology) for its predictive algorithm. In addition, they both require GRNs (not subpopulation specific) for their algorithms, although not as user's input. We have clarified this issue in the introduction and result of the revised manuscript. Regarding the comparison with ref. 32 and 33, this point was also discussed in the response to the last comment. However, we provide more detailed discussion of each method below.

CellNet (ref. 32): Although CellNet can accept user input data for the starting cell/tissue type, it compares it with bulk microarray data. For this reason, we could not use it for single-cell sequencing data, as it was technically difficult to standardize between single-cell sequencing and bulk micorarray data and addressing this issue is beyond the scope of this study. In addition, the prediction of reprogramming factors in CellNet is currently limited to only 20 built-in target cell/tissue types, no combination of which corresponded to combinations among different subpopulations in the single-cell data we collected in this study.

Mogrify (ref. 33): Currently, Mogrify has a fixed panel of starting and end cell/tissue types, for which it can make predictions. Further, it does not accept user input gene expression data. Therefore, for a direct comparison with Mogrify performance, we could only use those cell/tissue types that were also present in the single-cell datasets we used in this study. Since the subpopulations in our single-cell data compendium shared a very limited overlap with the cell/tissue type present in Mogrify, we performed the

analysis for this limited set of cell/tissue types alone (Table S6). The result showed that Mogrify identified the reprogramming factors for ESCs (or iPSCs) correctly. However, its predictions for neurons (from NSCs), lung fibroblast and bronchial epithelial cells did not contain any known TFs.

Due to the above-mentioned reasons, a systematic comparison between our tool and the tools from ref. 32 and 33 is not possible. Nevertheless, we have mentioned this point in the result and discussion of the revised manuscript, and added the Mogrify result in Table S6.

For those methods for which a comparison is possible, it must be carried out as it has been for JSD, and for those where it is not possible (32?,33?), a more complete explanation must be given than that provided. The current explanation is flawed and it is not stated that comparison (e.g. to CellNet) is not possible. For a methods paper, this is essential.

This point was addressed in the response to the last comment. Please see above.

Apart from this isolation from previous work and lack of systematic comparison, the paper and the work is excellent, with very exciting experimental validation. Single-cell sequencing is certainly a topic of high general interest at the moment, and this paper could be well received.

We thank the reviewer for the positive comment on our work.

On the related topic of distinguishing this paper from other work, it would be interesting to hear more about differences using single-cell data, and important to understand what is fundamentally and intrinsically different about sub-populations of cells versus different cell types. The authors present the differences as fundamental, but it is not explained why. The authors claim a 'conceptual advancement' without discussion of how their concept is different from cell-type work that precedes it; it does look like application of an existing concept to single-cell data.

The first point was discussed in the response to the first comment of this reviewer. The important point is that the definition of identity is relative, and depends on the context. In general, when dealing with subtypes/subpopulations within a tissue or organ, the definition of identity becomes more fine than when comparing two different tissues. For example, although dopaminergic and GABAergic neurons will have distinct identities when they are compared to each other, they have certain common neuronal identities when compared to hepatocytes. Within this context, our current work aims to identify such fine differences among cell subpopulations within a tissue or organ. This part is now explained in the introduction of the manuscript.

Regarding the 2nd point, the conceptual advancement we put forth is that transcriptional synergy captures the identity of cell subpopulations. However, we do not claim that the concept of subpopulation identity itself is a major novelty of this work.

Reviewer #3 (Remarks to the Author):

Okawa et al. developed a computational platform (TransSyn) for identifying synergy of transcription factors in single cell RNA-seq data. The advantage of their platform is that it only requires a large set of single-cell RNA-seq data from distinct subpopulations as input and does not require pre-compiled reference datasets. The TransSyn can find some critical synergistic transcription factors, like NANOG, POU5F1, SOX2 for ESCs and Myt1l for neurons.

The authors demonstrated the superiority of their system in predicting the combination of TFs for cell identity when compared with JSD method. They should also compare with a newly developed method developed by Rackham et al (Nature Genetics, 2016) even though that method was developed to predict core TFs for transdifferentiation.

This point was also raised by the 2nd reviewer and we extensively addressed it, including the comparison with Mogrify (by Rackham et al.). Basically, currently Mogrify has fixed starting and end cell/tissue types, for which it can make predictions and it does not accept user input gene expression data. For this reason, for a direct comparison with Mogrify, we could only use those cell/tissue types that were present both in single-cell datasets we used in this study and in Mogrify. Since the subpopulations in our single-cell data compendium shared a very limited overlap with the cell/tissue type present in Mogrify, we performed the analysis for this limited set of cell/tissue types alone (Table S6). The result showed that Mogrify identified the reprogramming factors for ESCs (or iPSCs) correctly. However, its predictions for neurons (from NSCs), lung fibroblast and bronchial epithelial cells did not contain any known TFs. We have mentioned this point in the result and discussion of the revised manuscript, and added the Mogrify result in Table S6.

The example for the application of the synergistic transcription factors is not convincing. There is no data showing that FGF-EGF expanded progenitors hNES exhibit the hindbrain identity. So, the rationale behind the use of DKK1 to switch to the midbrain fate is not clear. No quantification of TH+ neurons and other DA neuron markers, such as NR4A2 and PITX3v are provided to support their claim of "more efficient differentiation towards dopaminergic neurons than standard protocols". Additional proof by directing the cell fate by transgenic expression of transcription factors OTX2, LMX1A, FOXA2 is a plus.

In the present study we decided not to use forced expression of the identity genes (i.e reprogramming) because this strategy would limit our capacity to use these genes to evaluate the acquisition of the correct cell identity. Instead, we chose to use morphogens and small molecules to change the identity of the NES cells and induce their differentiation. We think this is a better option to evaluate the correct cell identity, as it can be easily done by analyzing the expression of the endogenous identity TFs. Another reason to follow this approach is that morphogens and small molecules are the current tools of choice for the development of stem cell differentiation protocols aiming at generating midbrain dopaminergic neurons for transplantation in PD. In addition, the manipulations performed in this study directly activate the signaling pathways controlling cell identity. SAG (Smoothed agonist) is known to activate the Shh receptor smoothed (Chen et al. 2002) and induce FOXA2 (Denham et al., 2012), which is required for floor plate development (Sasaki and Hogan 1994). On the other hand, Dkk1 directly modulates Wnt signaling (Glinka et al., 1998) to reach the levels required to induce OTX2 (Chung, et al. 2009) and for midbrain dopamine neuron development (Ribeiro et al., 2011). Our data also shows that treatment of hNES cells with SAG+ Dkk1 results in the expected increases in FOXA2:GBX2 ratio and OTX2 (Figure 2), supporting the notion that the compounds we used target the TFs predicted by TransSyn to define hProgFPM identity. This is now more explained in the result section.

In the revised version of the manuscript we now make clear that the NES cells (SAI2 cell line) were isolated from the developing hindbrain and in addition of their identity TFs (Fig 2A) they express hindbrain markers such as GBX2, but not midbrain markers such as OTX2 (Tailor et al., 2013). We also explain that

the rationale for the use of Dkk1 is that hindbrain progenitors are exposed to higher concentrations of Wnt compared to midbrain progenitors during development. We reasoned that by using the Wnt antagonist, Dkk1, hindbrain progenitors could be posteriorized and converted to midbrain progenitors. Our results show that a 2 day Dkk1 and SAG treatment induced ventral midbrain regional identity, as shown by the increase in FOXA2 expression (Fig. 2F), the increase in the ratio OTX2:GBX2 (Fig 2E) and the increase in the number of OTX2+ cells after 2 days of treatment (Fig 2I).

To further examine the identity of the converted cells and their functionality as converted midbrain progenitor cells, we examined their capacity to: (i) express more differentiated markers such as Lmx1a (appearing after OTX2 and FOXA2 in development, see Arenas et al., 2015 for review); and (ii) differentiate into midbrain dopaminergic neurons. For this purpose we used a protocol in which NES cells are treated for 2 days with Dkk1 and SAG (conversion) and differentiated for 6 days (Fig 3A). To determine whether midbrain patterning with SAG and Dkk1 was efficient or could be further enhanced, we compared the standard protocol with a protocol using FGF8, a factor that has been recently reported to improve midbrain patterning and differentiation (Kirkeby et al., 2017). Our results show that while both protocols further increased OTX2 and decreased GBX2 expression, the addition of FGF8 did not significantly increase the expression of LMX1A, a critical TF for midbrain dopaminergic differentiation (Andersson et al., 2006). In addition while both protocols increased midbrain dopamine markers such as NR4A2 and DAT, only SAG and Dkk1 treatment (in the absence of FGF8) significantly increased (by 5-fold) the number of TH+ cells. These results indicate that FGF8 is not required to increase the overall efficiency of our protocol and suggested that Dkk1 and SAG are sufficient to change the identity of NES cells.

To further confirm these results, we performed immunocytochemistry at day 8 of differentiation and found rare, immature and very weak TH+ cells in the control condition and a very strong increase in the number and morphological complexity of TH+ cells after SAG and Dkk1 treatment (Fig 3D,E). Moreover, this increase was accompanied by an increase in the number of OTX2+ cells (Fig 3D,F). Finally we also found that TH+ cells in the SAG+ Dkk1 condition co-express the midbrain markers NURR1 and LMX1A. Combined, our results of the analysis of TF expression and functionality after conversion and differentiation of NES cells, shown in the new Figure 3 of the ms, confirm that SAG and Dkk1 treatment induce the expression of the identity TFs identified by TranSyn and, as predicted, convert NES cells into medial midbrain floor-plate progenitors, capable of generating dopamine neurons.

References:

- Mazzoni EO,, Mahony S, Closser M, Morrison CA, Nedelec S, Williams DJ, An D, Gifford DK, Wichterle H. Synergistic binding of transcription factors to cell-specific enhancers programs motor neuron identity. *16*, 1219–1227 (2013). doi:10.1038/nn.3467 pmid:23872598
- Andersson E, Tryggvason U, Deng Q, Friling S, Alekseenko Z, Robert B, Perlmann T, Ericson J. Identification of intrinsic determinants of midbrain dopamine neurons. *Cell*. 2006 Jan 27;124(2):393-405
- Chen JK, Taipale J, Young KE, Maiti T, Beachy PA. Small molecule modulation of Smoothed activity. *Proc. Natl Acad. Sci. USA* 2002 **99**, 14071–14076
- Chung S, Leung A, Han BS, Chang MY, Moon JI, Kim CH, Hong S, Pruszk J, Isacson O, Kim KS. (2009). Wnt1-lmx1a forms a novel autoregulatory loop and controls midbrain dopaminergic differentiation synergistically with the SHH-FoxA2 pathway. *Cell Stem Cell* 5, 646-658.
- Denham M, Bye C, Leung J, Conley BJ, Thompson LH, Dottori M. (2012) Glycogen synthase kinase 3 β and activin/nodal inhibition in human embryonic stem cells induces a pre-neuroepithelial state that is required for specification to a floor plate cell lineage. *Stem Cells* 30(11):2400-11.

Glinka A, Wu W, Delius H, Monaghan AP, Blumenstock C, Niehrs C. (1998) Dickkopf-1 is a member of a new family of secreted proteins and functions in head induction. *Nature* 391(6665):357-62.

Kirkeby A, Nolbrant S, Tiklova K, Heuer A, Kee N, Cardoso T, Ottosson DR, Lelos MJ, Rifes P, Dunnett SB, Grealish S, Perlmann T, Parmar M. Predictive Markers Guide Differentiation to Improve Graft Outcome in Clinical Translation of hESC-Based Therapy for Parkinson's Disease. *Cell Stem Cell*. 2017 Jan 5;20(1):135-148.

Ribeiro D, Ellwanger K, Glasgow D, Theofilopoulos S, Corsini NS, Martin-Villalba A, Niehrs C, Arenas E. (2011) Dkk1 regulates ventral midbrain dopaminergic differentiation and morphogenesis. *PLoS One*. 6(2):e15786.

Sasaki H, Hogan BL. (1994) HNF-3 beta as a regulator of floor plate development. *Cell* 76(1):103-15.

Reviewers' comments:

Reviewer #1 (Remarks to the Author):

The manuscript has been much improved, in particular with respect to the experimental validation provided. The authors now also present an excellent rationale for why they didn't directly overexpress transcription factors. Overall, TransSyn represents a valuable contribution to the field of cell engineering.

Reviewer #2 (Remarks to the Author):

What the authors mean by sub-populations is now discussed in the introduction but remains vague with illustration by example. According to the introduction, occurring within the same organ is sufficient to be described as a sub-population. Even different cell types within the same tissue are most commonly regarded as distinct cell types and not subpopulations of the same type. I would imagine for example that most readers would imagine dopaminergic and GABAergic neurons are a different cell type, whereas the authors describe them as subpopulations in the rebuttal. The authors seem to include even more different cell types in their usage of subpopulations; basically the definition of subpopulations versus primary cell type for the reader in the introduction is still inadequate.

The authors have removed some inaccurate statements but have introduced another. The Mogrify method from Rackham et al. also claims to take into account transcriptional synergy via "combined influence".

The context of previous methods in the field (even if not directly comparable) is much improved with comparisons made.

The additional comparison and discussion of CellNet and Mogrify in S6 is welcome but would be improved if the factors identified from the publications were listed in another column for comparison, especially for those where the factors were not predicted, and there is no pubmed reference to Guo et al. 2015.

The authors still do not clarify how the concept of transcriptional synergy as a conceptual advancement is different from the "combined influence" described in the Rackham et al. paper.

Reviewer #3 (Remarks to the Author):

The authors made some effort to demonstrate that the hindbrain progenitors can be converted to midbrain progenitors which give rise to TH+ dopamine neurons. However, the population of TH+ dopamine neurons is less than 0.3% even after treatment with SAG+DKK1. Such a subtle effect can be contributed to many factors, including the preferential survival/expansion of midbrain progenitors by SAG and/or relative inhibition of hindbrain progenitors by DKK1. Given that this is the only confirmation experiment at the cellular level to validate the TransSyn program, it remains a question if the program can indeed predict the transcription code for a subpopulation of cells.

We would like to thank again all the reviewers for constructive comments. Please see below our response to each reviewers comment in blue. All major changes in the revised manuscript are highlighted in yellow.

Dear Dr del Sol,

Your manuscript entitled "Transcriptional synergy as an emergent property defining cell subpopulation identity and its application to population shift" has now been seen again by our 3 referees. You will see from their comments below that while they are satisfied with most of your revisions, Reviewer 3 continues to raise concerns regarding your validation experiments. We are interested in the possibility of publishing your study in Nature Communications, but would like to consider your response to these concerns in the form of a revised manuscript before we make a final decision on publication.

In light of the comments of Reviewer 3 we sought addition advice from Reviewer 1, who does not fully share Reviewer 3's concern about the efficiency of hNES conversion to hPRogFPM and their differentiation into DApepic neurons. However, Reviewer 1 recommends that additional analyses be performed to further support the validation presented in Figure 3. Reviewer 1 recommends additional immunostaining and quantification to assess the cellular changes being described in the treated cells, and more repeats of the experiments to boost confidence in the results. We would recommend additional immunostaining experiments to show the neuronal morphology of the differentiated cells (e.g. synaptic and/or axonal markers) to buttress the existing data on TH+.

Following the suggestions of the editor and Reviewer 1, we have performed 3 more differentiation experiments and confirmed that TH+ cells are nearly completely absent in the control condition and readily observed in the SAG+ Dkk1 condition. In addition we have stained converted and differentiated cultures for one additional midbrain marker, Pbx1, a TF required for midbrain development (Villaescusa et al., 2016). We show that TH+ cells in our culture are also Pbx1+ (Figure 3I), suggesting that they have a correct midbrain identity. We have also examined the degree of maturation of the TH+ cells and we found that, despite the fact that they were treated for only 8 days, they expressed the mature neuronal marker MAP2 (appearing in hESC cultures only at day 21), most of the TH+ cells were bipolar and some of them exhibited very long processes with varicosities (Figure 3J and K), a typical morphology of mDA neurons. These results show that TH+ cells in our cultures can achieve a relatively high degree of neuronal maturation despite the short differentiation time. This data has been included in the revised version of the manuscript.

Reviewer 1 also noted to us that the scales of the images appear to differ between the two conditions in Fig. 3D, which we would ask you to clarify. Please also note that Reviewer 2 also has a few remaining concerns, which should be addressed.

As indicated by Reviewer 1, we have verified the magnifications in Figure 3D and found that the scale is the same in the two conditions. Our results show a difference in the size and shape of the nuclei that we think reflects the distinct degree of differentiation in control versus SAG+Dkk1, as

indicated by the increase in differentiated markers in Figures 3B-C.

Reviewers' comments:

Reviewer #1 (Remarks to the Author):

The manuscript has been much improved, in particular with respect to the experimental validation provided. The authors now also present an excellent rationale for why they didn't directly overexpress transcription factors. Overall, TransSyn represents a valuable contribution to the field of cell engineering.

Thank you for your positive comment.

Reviewer #2 (Remarks to the Author):

What the authors mean by sub-populations is now discussed in the introduction but remains vague with illustration by example. According to the introduction, occurring within the same organ is sufficient to be described as a sub-population. Even different cell types within the same tissue are most commonly regarded as distinct cell types and not subpopulations of the same type. I would imagine for example that most readers would imagine dopaminergic and GABAergic neurons are a different cell type, whereas the authors describe them as subpopulations in the rebuttal. The authors seem to include even more different cell types in their usage of subpopulations; basically the definition of subpopulations versus primary cell type for the reader in the introduction is still inadequate.

We would like to clarify that our definition of subpopulation is, simply put, distinct groups of cells within a given heterogeneous population of cells, which are identified by unbiased- (e.g., clustering, PCA) and/or biased means (e.g., marker expressions) based on the transcriptomics profile. In this regard, our definition of subpopulation entails “cell types”, as suggested by the reviewer and we admit that the example of dopaminergic and GABAergic neurons was not a good one. Nevertheless, we prefer to use the term “subpopulation” to “cell type”, as our examples include not only well-defined cell types but also subtypes of a same cell type (e.g., dopaminergic neurons 0, 1 and 2, active/quiescent NSCs 1 and 2, etc.) and also those whose identity is indeed unclear (e.g., unknown enterocytes 1 and 2, undefined pancreatic cells, etc.). In this study we refer all these cases as “subpopulation”. This point is now more clarified in the introduction and result of the revised manuscript.

The authors have removed some inaccurate statements but have introduced another. The Mogrify method from Rackham et al. also claims to take into account transcriptional synergy via "combined influence".

What Rackham et al. means by “combined influence” is to identify a set of TFs that can reach as many target genes as possible in a given GRN in order to maximize reprogramming efficiency. In this regard, the kind of synergy mentioned by Rackham et al. considers the individual property of each TF (i.e, how many targets it can reach) and their additive effect, without considering potential interactions (direct or indirect) among the candidate TFs themselves. On the other hand, our definition of synergy is among potential identity TFs themselves measured by a mathematically well-defined information theoretic measure of synergy, which is not based on individual properties of TFs (e.g., the number of target genes). Nevertheless, it is true that the “combined influence” could be considered a type of synergy in a broad sense. Therefore, we made this point clearer in the

revised manuscript.

The context of previous methods in the field (even if not directly comparable) is much improved with comparisons made.

Thank you.

The additional comparison and discussion of CellNet and Mogrify in S6 is welcome but would be improved if the factors identified from the publications were listed in another column for comparison, especially for those where the factors were not predicted, and there is no pubmed reference to Guo et al. 2015.

There is no literature evidence that shows that these factors can maintain the identity of the respective subpopulations in Guo et al. 2015 dataset. Therefore, the Pubmed column was left empty.

The authors still do not clarify how the concept of transcriptional synergy as a conceptual advancement is different from the "combined influence" described in the Rackham et al. paper.

The intrinsic difference between the "combined influence" and our synergy is discussed in the 2nd point above. However, we agree that a "conceptual advancement" is rather an unclear statement. Thus, we also rephrased it as "novel approach" in the conclusion.

Reviewer #3 (Remarks to the Author):

The authors made some effort to demonstrate that the hindbrain progenitors can be converted to midbrain progenitors which give rise to TH+ dopamine neurons. However, the population of TH+ dopamine neurons is less than 0.3% even after treatment with SAG+DKK1. Such a subtle effect can be contributed to many factors, including the preferential survival/expansion of midbrain progenitors by SAG and/or relative inhibition of hindbrain progenitors by DKK1. Given that this is the only confirmation experiment at the cellular level to validate the TransSyn program, it remains a question if the program can indeed predict the transcription code for a subpopulation of cells.

Our experiments were performed with a neuroepithelial cell *isolated from the human hindbrain* (SAI2) that was verified to express typical hindbrain markers and had a hindbrain identity (Taylor et al., 2013). Our results in the control condition (Figures 2 and 3) also confirm the hindbrain identity of the cells, as previously published. We think it is highly unlikely (or impossible) that a midbrain progenitor could have been isolated from the hindbrain region and remained undetected in previous and current analyses and is now expanded or supported in our preparation. Our results clearly indicate that Dkk1 (in combination with SAG) does not only inhibit hindbrain differentiation, as the reviewer also suggests, but it additionally instructs a midbrain identity. This is clearly shown by the acquisition of the midbrain markers shown in Figure 2, just by Dkk1 and SAG treatment for 2 days, and the capacity of the converted cells to differentiate into mDA neurons, as shown in figure 3. We thus conclude that SAI2, a cell type with lateral hindbrain origin and identity, is converted into cells with a ventral midbrain phenotype by exposure to SAG and Dkk1.

To further reinforce our observation, in the revised version of the manuscript we have performed 3 more differentiation experiments and confirmed, as already shown in Figure 3F a near complete absence of TH+ cells in the control condition and a clear and significant presence of TH+ cells in the SAG+ Dkk1 condition. These cultures have been stained with an additional marker, Pbx1, a TF that is present in TH- neuroblasts and is required for their differentiation into TH+ mDA neurons (Villaescusa et al., 2016). Consistent with these results, we found that Pbx1 is present in both TH- and TH+ cells in our cultures, suggesting that cells have midbrain identity and are

differentiating into mDA neurons. We have also examined the degree of maturation of the TH+ cells in our cultures and we found that although cells were cultivated for only 8 days, they expressed the mature neuronal marker MAP2 (Figure 3J), a marker that appears in hESC cell cultures only at day 21. In addition, we observed that most of the TH+ cells have a bipolar morphology and that some of them exhibited very long processes with varicosities (Figure 3J and K), a typical morphology of mDA neurons. These results show that TH+ cells in our cultures can achieve a relatively high degree of morphological maturation in a relatively short differentiation time.

REVIEWERS' COMMENTS:

Reviewer #1 (Remarks to the Author):

The authors have satisfied all the concerns I had previously raised. Moreover, I feel that the functional studies provided support the author's approach.